# Identification of Differentially Expressed Genes and Pathways for Abdominal Fat Deposition in Ovariectomized and Sham-Operated Chickens

**DOI:** 10.3390/genes10020155

**Published:** 2019-02-18

**Authors:** Xiaopeng Mu, Xiaoyan Cui, Ranran Liu, Qinghe Li, Maiqing Zheng, Guiping Zhao, Changrong Ge, Jie Wen, Yaodong Hu, Huanxian Cui

**Affiliations:** 1Farm Animal Genetic Resources Exploration and Innovation Key Laboratory of Sichuan Province, Sichuan Agricultural University, Chengdu Campus, Chengdu 611130, China; xiaopengmu07@126.com; 2Institute of Animal Sciences, Chinese Academy of Agricultural Sciences, Beijing 100193, China; cxyan813@163.com (X.C.); liuranran112@126.com (R.L.); qli2014@126.com (Q.L.); wheatzheng@163.com (M.Z.); zhaoguiping@caas.cn (G.Z.); wenjie@caas.cn (J.W.); 3State Key Laboratory of Animal Nutrition, Beijing 100193, China; 4Institute of Animal Science, Guangdong Academy of Agricultural Sciences, State Key Laboratory of Livestock and Poultry Breeding, Key Laboratory of Animal Nutrition and Feed Science in South China of Ministry of Agriculture, Guangdong Public Laboratory of Animal Breeding and Nutrition, Guangdong Key Laboratory of Animal Breeding and Nutrition, Guangzhou 510640, China; 5Yunnan Provincial Key Laboratory of Animal Nutrition and Feed, Yunnan Agricultural University, Kunming 650201, China; gcrzal@126.com

**Keywords:** chicken, abdominal fat, steroid biosynthesis, gene expression profiling, differentially expressed genes, glycerolipid metabolism

## Abstract

Ovariectomy results in improved meat quality (growth rate, tenderness, and flavor) of broilers. However, some negative effects increased (abdominal fat (AF) deposition, low feed conversion, etc.) have also been reported. In this study, the gene expression profiles of AF tissue in ovariectomized and sham-operated chickens were determined to identify differentially expressed genes (DEGs) and pathways to explore the molecular mechanisms underlying AF accumulation. Comparing the ovariectomized group and the sham-operated group, the abdominal fat weight (AFW) and abdominal fat percentage (AFP) were increased significantly (*p* < 0.05) at 14 and 19 weeks after ovariectomy. According to the gene expression profiling analysis, 108 DEGs of fat metabolism were screened from 1461 DEGs. Among them, *ABCA1*, *ABCACA*, *LPL*, *CREB1*, *PNPLA2*, which are involved in glycerolipid—or steroid—associated biological processes, and the hormone receptor genes, *ESR1* and *PRLR*, were down-regulated significantly in the ovariectomized group compared to the sham-operated group (*p* < 0.05). Conversely, *CETP*, *DGAT2*, *DHCR24*, *HSD17B7* and *MSMO1*, were significantly up-regulated (*p* < 0.05) after ovariectomy. Based on the DEGs, the glycerolipid metabolism, steroid biosynthesis, and other signaling pathways (MAPK, TGF-β, and adhesion pathways, etc.) were enriched, which may also contribute to the regulation of AF deposition. Our data suggest that AF deposition was significantly increased in ovariectomized chickens by the down-regulation of the decomposition genes of glycerolipid metabolism, which inhibits AF degradation, and the up-regulation of steroid biosynthesis genes, which increases fat accumulation. These findings provide new insights into the molecular mechanisms of fat deposition in the ovariectomized chickens.

## 1. Introduction

Ovariectomy is the removal of the ovaries of a female animal using a technique known as ovariectomization. After the ovariectomy, production performance, slaughter performance, blood biochemistry, fat metabolism, bone development, immunity, and other characteristics can undergo significant changes [1,2], which is widely used in production. For example, a previous study showed that abdominal fat (AF) deposition increases significantly after ovariectomy or caponization [3]. In general, fat deposition occurs in subcutaneous tissues, muscles, and the abdomen in chickens. The AF tissue is a reliable parameter for judging total body fat content because it is linked directly to total body fat content in avian species [4,5] It is crucial in poultry because it grows faster compared with other fat tissues [6], and has a higher fat content, accounting for approximately 30% of the mass of major fat depots [7]. Reducing AF content is an urgent problem to be solved.

Adipocytes are derived from mesenchymal stem cells, which differentiate into adipoblasts, preadipocytes, and mature adipocytes. After birth, AF deposition is increased not by an increase in the number of adipocytes, but by an increase in the volume [8]. The main sources of fat in chickens are the intestinal absorption of fat from food and the de novo synthesis in the liver [9]. As energy storage in the body, a large amount of biosynthesized or absorbed lipids is transported and stored as AF, which eventually leads to degradation in meat quality as fat accumulation becomes extreme.

Although there are some available studies on the regulatory mechanisms of AF deposition in chickens of different breeds or ages [7], contrastive analyses between ovariectomized and sham-operated chickens are scarce. The purpose of this study was to explore the molecular regulatory mechanism of AF deposition in Beijing-You chickens at 14 weeks after ovariectomy. Gene expression profiling was used to identify candidate genes and related signaling pathways. The present findings provide a theoretical basis for producing higher quality chicken meat and for improving production efficiency.

## 2. Materials and Methods

### 2.1. Ethics Statement

This study was conducted in accordance with the Guidelines for Experimental Animals established by the Ministry of Science and Technology (Beijing, China). The protocol was approved by the Science Research Department (in charge of animal welfare issues) of the Institute of Animal Sciences (1 December 2013), Chinese Academy of Agricultural Sciences (Beijing, China) (No. IAS20131201).

### 2.2. Animals and Treatments

The study was conducted in accordance with the guidelines for experimental animals developed by the Ministry of Science and Technology of China (Beijing, China). Animal experiments were approved by the Science Research Department (in charge of welfare issues) at the Institute of Animal Sciences, Chinese Academy of Agricultural Sciences (CAAS) (Beijing, China). The Beijing-You (BJ-Y) chicken is a unique local breed in China, with high flavor characteristics. In this study, 80 female BJ-Y chickens with the same genetic background and similar weight were randomly distributed into two groups. The ovariectomy procedure was performed in accordance with previously described method in one of the two groups at three weeks [10], while the remaining group underwent a sham-operation (sham-operated group). Feed and water were withheld for 12 h before the operations. The incision site was sterilized with veterinary external disinfectant. A one centimeter lateral incision was made at the second-to-last rib, then, the ovaries were removed aseptically. The birds were raised in an environmentally-controlled room, in three-story step cages. Basal diets were formulated based on the National Resource Council (1994) requirements and the Feeding Standards of Chickens established by the Ministry of Agriculture, Beijing, China (2004). 

Under carbon dioxide anesthesia, 10, 20, 10 chickens were euthanized by severing the carotid artery at 10, 14, 19 weeks after the ovariectomy or sham-operations in each group, respectively. After slaughter, the AF was dissected in the same area for all chickens. The AF samples were weighed, snap-frozen in liquid nitrogen, and stored at −80 °C until use. The remaining AF tissues were removed and weighed, which included the AF pad and fat around the gizzard. The AF weight (AFW) was recorded, and the AF percentage (AFP) was calculated (mass of AF as a percentage of the live weight). 

### 2.3. RNA Extraction and Identification 

Total RNA was isolated from AF tissues of chickens, at 14 weeks after ovariectomy or sham-operation, using the Trizol reagent (DP419) according to the manufacturer’s protocol. The quality of RNA was detected by 1% gel electrophoresis, and RNA concentration was determined by NANODROP2000 spectrophotometer (Thermo Scientific, Hudson, DE, USA). The OD260/280 values of all samples were limited to the range of 1.8 to 2.0. The RNA samples were subsequently used for gene expression profiling.

### 2.4. Gene Expression Profiling and Data Analysis

Based on ultra-high-throughput sequencing (HiSeq2500; Illumina, San Diego, CA, USA), gene expression profiling was undertaken by Berry Genomics (Beijing, China). Raw data were converted to FASTQ files using bcl2fastq (Illumina). Clean reads were generated by removing reads with adapter and low-quality sequences and mapped to the reference chicken genome and genes (Gallus gallus, Galgal4; available at https://www.ncbi.nlm.nih.gov/assembly/GCF_000002315.3) using TopHat 1.3.2 (https://ccb.jhu.edu/software/tophat). Gene expression levels were calculated using the RPKM method, as described by Mortazavi et al. [11]. Differentially expressed genes (DEGs) between the ovariectomized and sham-operated groups were analyzed using the edgeR R package. DEGs were screened by the following criteria: |log2 FC| ≥ 0.58, with *p* < 0.05. Based on the DEGs, clustering analysis was performed in each sample by the pheatmap package of R software. Hierarchical clustering was performed on both rows and columns, and the resulting dendrogram was saved as an image file. GO (Gene Ontology) enrichment analysis was performed to identify the gene function classes and categories corresponding to the DEGs using the ClueGO plug-in and CluePedia plugin of Cytoscape (https://cytoscape.org/). The significance level of GO terms enrichment was set at *p* < 0.05 as indicated in the Yekutieli method [12]. According to the results of GO enrichment analysis, DEGs related to abdominal adipose tissue metabolism were screened. The significantly enriched signaling pathways of DEGs were analyzed by the KEGG (Kyoto Encyclopedia of Genes and Genomes) by Kobas3.0 [13]. *p* < 0.05 was considered to be indicative of statistical significance. 

### 2.5. qRT-PCR Analysis 

Using the same RNA samples, real-time quantitative polymerase chain reaction (qRT-PCR) was performed to confirm the results of gene expression profiling. RNA samples were reverse transcribed using TIANGEN^®^ FastQuant RT Kit (Tiangen, Beijing, China), and specific primers were designed placing at or just outside of exon/exon junctions using Primer 5.0 software dependent on GeneBank sequences (Appendix A).

Samples were amplified using the real-time PCR Detection System ABI 7500 (Applied Biosystems, Shanghai, China). The PCR mixture contained 10 μL of 2 × iQ™ SYBR Green Supermix, 0.5 μL (10 mmol) of each primer, and 1 μL of cDNA, along with ddH2O for a total volume of 20 μL. After initial denaturation for 30 s at 95 °C, amplification was performed for 40 cycles (95 °C for 5 s and 60 °C for 32 s). PCR efficiency for these genes and β-actin was consistent. The comparative cycle threshold (CT) method was used to determine fold-changes in gene expression [14], with fold-changes being calculated as 2^−ΔΔCT^. The results were expressed as the mean fold-change in gene expression from triplicate analyses, using the sham-operated group samples as the calibrators (arbitrarily assigned an expression level of 1 for each gene). Correlations between relative abundance from qRT-PCR and gene expression profiling data were also calculated. 

### 2.6. Statistical Analysis

Statistical differences between groups were evaluated using the Student’s *t*-test. All computations were made suing SPSS Version 20.0 (SPSS Inc. Chicago, IL, USA). *p* < 0.05 was considered significant, and data are presented as mean ± SEM.

## 3. Results

### 3.1. Ovariectomy Accelerates AF Deposition

Data on the AFW and AFP of ovariectomized and sham-operated chickens at 10, 14, and 19 weeks after ovariectomy or sham-operations are depicted in Figure 1. The AFW and AFP of the ovariectomized group (40.83 g, 3.96%) were significantly higher (*p* < 0.05) than those (27.1 g, 2.95%) of the sham-operated group at 14 weeks. Similar observations were recorded at 19 weeks post-surgery (74.53 g, 6.58% vs. 39.92 g, 3.92%; *p* < 0.05). There was more AF deposition in the ovariectomized group than in the sham-operated group, which suggests that AF deposition could increase after ovariectomy.

### 3.2. Identification of DEGs Related to AF Metabolism After Ovariectomy

Three representative chickens per group were selected at 14 weeks after ovariectomy or sham-operations, and RNA was extracted from AF tissues. Using gene expression profiling and comparing the ovariectomized group with the sham-operated group (Figure 2a), a total of 1461 DEGs (|log2 FC| ≥ 0.58, with *p* < 0.05) were screened, of which 1009 were down-regulated and 452 were up-regulated (Appendix A). Based on these 1461 DEGs, cluster analysis was performed, and the results showed that samples of the same groups were clustered together, respectively (Figure 2b).

GO analysis was performed based on 1461 known DEGs, and the main GO terms included the following processes: Cell adhesion, regulation of the immune system, regulation of cell differentiation, transmembrane transport, and lipid metabolism (Appendix A). GO enrichment analysis indicated that a large number of DEGs related to lipid metabolism (Appendix A), and 12 representative DEGs were selected to validate the gene expression profiling results by qRT-PCR (Figure 3). Among them, the cholesterol homeostasis genes *ABCA1*, *ABCACA*, *LPL* and *CREB1* were down-regulated, but *CETP* was significantly up-regulated; glycerolipid metabolism gene *PNPLA2* was significantly down-regulated, but *DGAT2* was significantly up-regulated; steroid biosynthesis genes *MSMO1*, *HSD17B7* and *DHCR24* were significantly up-regulated; and hormone receptor genes *ESR1* and *PRLR* were also significantly down-regulated after ovariectomy (*p* < 0.05). To confirm the accuracy of the data, the results of gene expression profiling and qRT-PCR were analyzed by Spearman rank correlation, and the fold change of gene expression between the two methods was significantly correlated (Figure 4) (*r* = 0.9802, *p* < 0.05).

### 3.3. Pathways and Regulatory Network for AF in Chickens After Ovariectomy

Based on 1461 known DEGs, a KEGG pathway analysis was performed and 17 pathways were found to be significantly enriched (*p* < 0.05) (Figure 5). As expected, some pathways related to lipid metabolism were screened (e.g., glycerolipid metabolism, steroid biosynthesis, TGF-signaling pathway, MAPK signaling pathways, etc.). In addition, some cell adhesion signaling pathways (focal adhesion, adherens junction and regulation of actin cytoskeleton) were also significantly enriched (*p* < 0.05).

## 4. Discussion

### 4.1. Differences in AF Deposition between the Ovariectomized and Sham-Operated Groups 

Ovariectomy plays an important role in fat deposition. Studies have shown that the weight of AF in goats was significantly increased after ovariectomy [15]. AF deposition was also found to increase after ovariectomy in mice compared to a control group [16,17], and ovariectomy may stimulate adipocyte hypertrophy, therefore promoting obesity [18]. All these findings are consistent with our research which shows that AFW and AFP were up-regulated significantly in ovariectomized chickens at 14- and 19-weeks post-surgery, with the gap increasing over time. 

### 4.2. The Validation of Gene Expression Profiling and Identification of DEGs Related to AF Deposition

High quality total RNA samples were obtained and used for gene expression profiling. To validate the obtained results, cluster analysis was performed. The samples of the ovariectomized group and sham-operated group clustered closely, respectively. Furthermore, 12 representative genes were screened from 108 known DEGs related to fat metabolism, and qRT-PCR analysis was performed to validate the gene expression profiling. The fold changes in gene expression levels were used to correlate the results of RNA-seq and qRT-PCR, which revealed a significant correlation (*p* < 0.05). These results confirmed the reliability of the obtained gene expression profiling data.

### 4.3. Key Genes Related to the Metabolism of AF 

Based on the 1461 known DEGs, GO terms analysis was performed, the main GO terms included the following processes: Cell adhesion, regulation of the immune system, regulation of cell differentiation, transmembrane transport, and lipid metabolism. Among them, the gene expression levels of *ABCA1*, *ABCACA*, and *LPL* were significantly down-regulated in the ovariectomized compared to the sham-operated group. As a cholesterol efflux pump in the cellular lipid removal pathway, *ABCA1* has an important function in cholesterol efflux. Consequently, lower *ABCA1* expression could reduce lipid efflux, leading to lipid accumulation [19]. *ACACA* is important for malonyl CoA, which catalyzes the synthesis of long-chain fatty acids [20]. The *LPL* gene encodes lipoprotein lipase, which acts as the rate-limiting step in the hydrolysis of triglycerides, negatively correlating with fat deposition [21,22]. Similarly, *PNPLA2*, *CREB1*, *ESR1*, and *PRLR* had lower expression in AF of the ovariectomized group compared to the sham-operated group. *PNPLA2* encodes the key enzyme of triglyceride hydrolysis, participating in the regulation of the first step in triglyceride hydrolysis [23]. *CREB1* is up-regulated by P38 stimulation, and causes a decrease in steroidogenesis [24]. Estrogen are well-known regulators of fat distribution and deposition, with research showing lower *ESR1* expression in adipocytes of obese compared to non-obese premenopausal women [24,25]. *PRLR* has a positive contribution to triglyceride accumulation by increasing expression in the liver [25]. These results suggest that these genes could increase AF deposition by decreasing fat degradation.

Some genes (*MSMO1*, *DHCR24*, *HSD17B7*) that participate in the steroid biosynthesis process were up-regulated in the ovariectomized group compared to the sham-operated group. The *MSMO1* gene was expressed in diabetic animal models and shown to play key roles in cholesterol biosynthesis, energy metabolism, obesity, and dyslipidemia regulation [26,27]. *DHCR24* encodes the enzyme involved in the final step of cholesterol synthesis, catalyzing the synthesis of cholesterol from chain sterols [28]. *HSD17B7* encodes an enzyme that functions both as a 17-β-hydroxysteroid dehydrogenase in the biosynthesis of sex steroids and as a 3-ketosteroid reductase in the biosynthesis of cholesterol [29]. These results suggest that increasing expression of these genes could contribute to the increase in AF deposition after ovariectomy.

In addition, the expression levels of *DGAT2* and *CETP* were significantly higher in AF of the ovariectomized group than in the AF of the sham-operated group. *DGAT2* is critical for adipose tissue formation, catalyzing the final step in triacylglycerol synthesis. Overexpression of *DGAT2* in mammalian *HEK293* cells can significantly increase triglyceride synthesis [30,31]. *CETP* is involved in the transfer of neutral lipids (cholesteryl ester and triglycerides) and in the positively regulated phospholipid accumulation in chickens [32]. These results reveal that *DGAT2* and *CETP* could increase AF deposition in ovariectomized chickens. 

### 4.4. Signaling Pathways Related to Lipid Metabolism in AF after Ovariectomy

The regulation of AF deposition is possibly a result of complex interactions between multiple pathways. Based on the screened DEGs, several well-known pathways related to lipid metabolism were found to be enriched (i.e., glycerolipid metabolism, steroid biosynthesis, TGF-β signaling pathway, MAPK signaling pathway, etc.). In addition, some cell adhesion signaling pathways (focal adhesion, adherens junction and regulation of actin cytoskeleton) were also enriched significantly (*p* < 0.05), which was consistent with the results of a previous study [32].

Glycerolipid metabolism, regulated by *DGKH*, *DGKB*, *DGKE*, *DGKD*, *PNPLA2*, *PPAP2B*, *ALDH9A1*, *GPAM, LPL*, *AGPAT4*, *DGAT2*, has been reported to be significantly associated with fat metabolism [33]. According to previous research, *DGKE* plays a prominent role in enriching inositol phospholipids with unsaturated fatty acids and lipid synthesis [34]. *DGKD* knockout leads to a decrease in fatty acid synthesis-related enzymes, as well as in the levels of many lipids within the cell [35]. In addition, some steroid biosynthesis genes (*MSMO1*, *CYP51A1*, *HSD17B7*, *LIPA* and *DHCR24*) were also significantly up-regulated. As well known, the TGF-β and MAPK signaling pathways participate in the regulation of fat metabolism [36,37,38]. Interestingly, several studies suggest that cell adhesion-related pathways are involved in lipid deposition [3,38]. Three common DEGs (*VCL*, *ACTB* and *RHOA*) are shared by these pathways. Several DEGs of adhesion-related pathways (*FGF12*, *FGFR2*, *TGFBR2*) are involved in the MAPK signaling pathway [39]. Studies have shown that MAPK activation results in the disruption of tight junctions, whereas the inhibition of MAPK activation prevents this process [40]. Additionally, P38 was shown to negatively regulate steroidogenesis by increasing the expression of *CREB* [41]. These findings collectively suggest that multiple adhesion-related signaling pathways could interact and participate in the regulation of AF deposition.

Based on these pathways, a possible molecular regulatory network managing AF deposition in chickens was constructed (Figure 6), suggesting that the increased AF deposition may be the result of multiple signaling pathways acting together.

## 5. Conclusions

In this study, our data suggests that AF deposition was significantly increased in ovariectomized chickens by the down-regulation of the decomposition genes (*ABCA1*, *ABCACA*, *CREB1*, *PNPLA2*, *ESR1*, *PRLR*, *LPL*) of glycerolipid metabolism, which inhibits AF degradation, and the up-regulation of steroid biosynthesis genes (*CETP*, *DGAT2*, *DHCR24*, *HSD17B7*, *MSMO1*), which increases fat accumulation. They work together to increase the deposition of AF. Additionally, cell adhesions, TGF-β and MAPK signaling pathways were also found, which is possibly involved in the regulation of AF deposition. These findings provide new insights into the regulation of AF deposition in the ovariectomized chickens.

## Figures and Tables

**Figure 1 genes-10-00155-f001:**
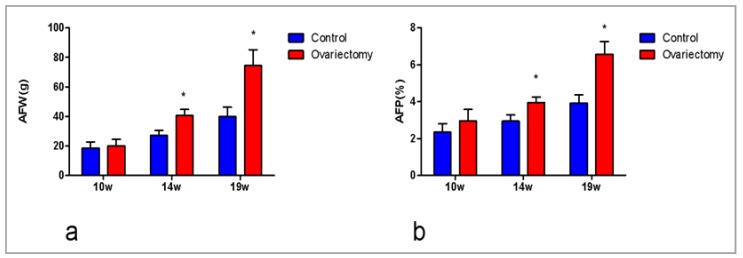
Abdominal fat weight (AFW) and abdominal fat percent (AFP) increase in the abdominal fat (AF) of female chickens after ovariectomy. AFW was increased in the AF of female chickens at 14 and 19 weeks after ovariectomy (**a**). AFP was increased in the AF of female chickens at 14 and 19 weeks after ovariectomy (**b**). Data are presented as means ± SEM (*n* = 20; * *p* < 0.05).

**Figure 2 genes-10-00155-f002:**
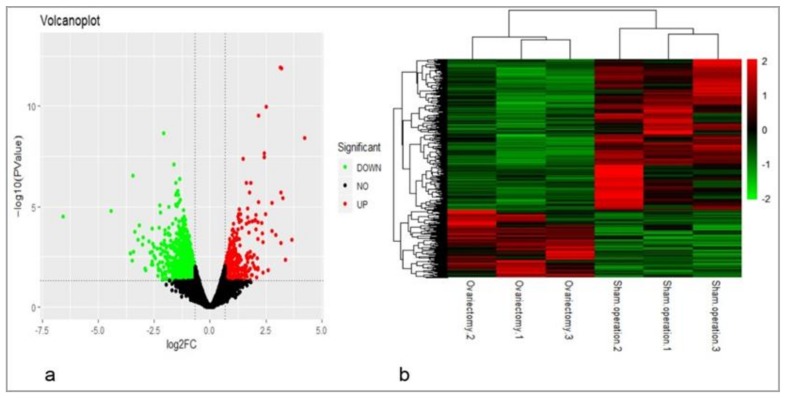
Volcano plot and cluster analysis of differentially expressed genes (DEGs). (**a**) Volcano plot. Red dots (UP) represent significantly up-regulated genes (log2 FC ≥ 0.58, *p* < 0.05); green dots (DOWN) represent significantly down-regulated genes (log2 FC ≤ −0.58, *p* < 0.05); black dots (NO) represent DEGs below the level of significance; (**b**) based on 1461 known DEGs in the AF tissue of the ovariectomized and sham-operated groups at 14 weeks post-surgery, cluster analysis was performed. The results show that the data in the gene expression profiling of chickens in same group were closely related.

**Figure 3 genes-10-00155-f003:**
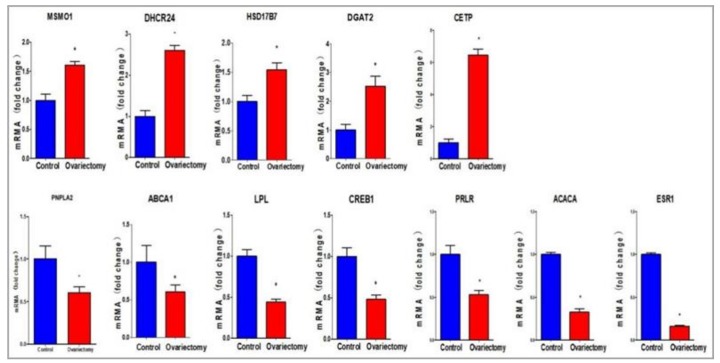
Real-time quantitative polymerase chain reaction (qRT-PCR) verification of DEGs detected by gene expression profiling. The expression levels of DEGs related to lipid metabolism determined by qRT-PCR in the ovariectomized and sham-operated groups. These genes were involved in glycerolipid metabolism, sterol biosynthesis, and fatty acid biosynthesis, among others. Each of these DEGs were up-regulated or down-regulated significantly (*p* < 0.05) in the ovariectomized group compared with the sham-operated group.

**Figure 4 genes-10-00155-f004:**
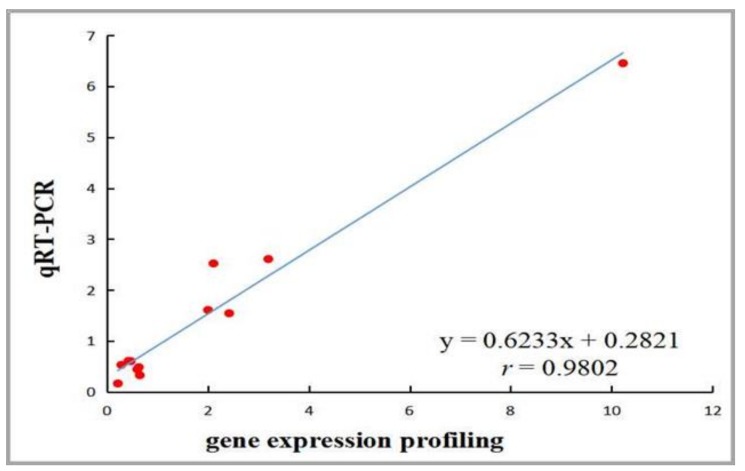
Correlation analysis of gene expression profiling and qRT-PCR results. The correlation between gene expression profiling and qRT-PCR data was analyzed by Spearman rank correlation in the ovariectomized and sham-operated groups. A high correlation coefficient (*r* = 0.9802, *p* < 0.05) was detected, which indicates that the gene expression profiling data are reliable.

**Figure 5 genes-10-00155-f005:**
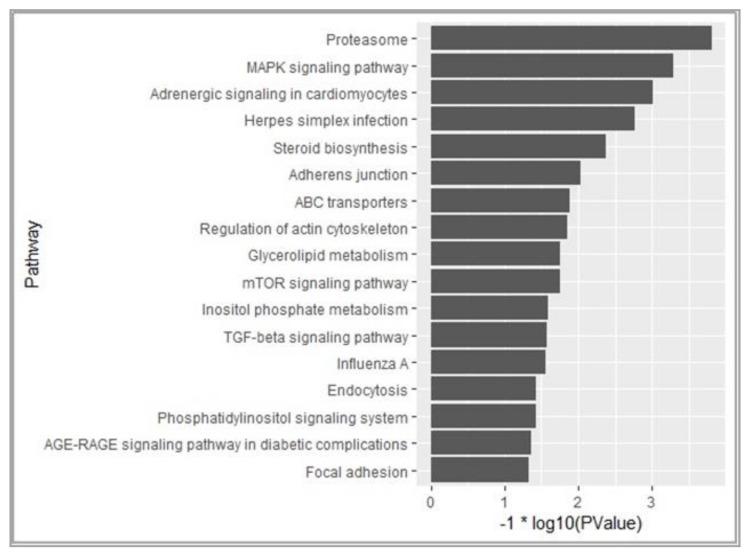
Enriched pathways based on the 1461 DEGs. KEGG (Kyoto Encyclopedia of Genes and Genomes) pathway analysis of DEGs showed that various fat metabolism pathways (glycerolipid metabolism, steroid biosynthesis etc.) were enriched (*p* < 0.05).

**Figure 6 genes-10-00155-f006:**
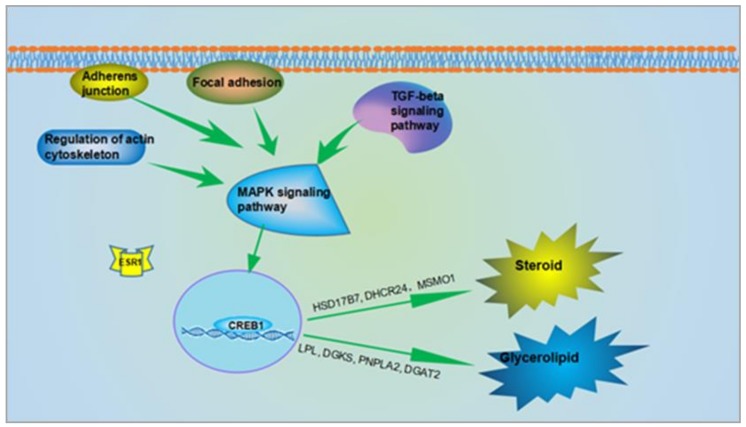
The potential regulatory network mediating lipid deposition in ovariectomized chickens. The AF deposition regulatory network may include: Glycerolipid metabolism, steroid metabolism, MAPK, TGF-β, and cell junctions (cell adhesion, adherens junctions, regulation of actin cytoskeleton) pathways.

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
