# Peer review of "Identification of Differentially Expressed Genes and Pathways for Abdominal Fat Deposition in Ovariectomized and Sham-Operated Chickens"

_genes, 2019, doi:10.3390/genes10020155_

Round 1

Reviewer 1 Report

The manuscript by Mu et al described a study on gene expression in a local breed of chickens after ovariectomy. The authors reported that ovariectomy resulted in an increased abdominal fat deposition, accompanied by differential regulation of a set of genes at the level of transcript abundance. 

Specific comments:

The authors used P < 0.05 as a cut-off for significantly differential expression change. Was this P-value corrected for multiple test? 

The age at which ovariectomy was performed should be clearly indicated. 

Line 178 is confusing, screening for what?

Figure 4. Are the RT-PCR data from the same samples that were used for RNA-seq analysis?

(Line 234) to the than?

What was the basis for the membrane localization of ESR1? 

Author Response

Manuscript Number: genes-410827

Identification of differentially expressed genes and pathways for abdominal fat deposition in ovariectomized and sham-operated chickens

Dear editor, 
We are grateful to reviewers for their effort, comments and valuable suggestions. We have carefully considered the comments and made several, appropriate changes to the text, point-by-point. References have been updated as well. We are confident that this revised version is superior to the original. Here are details of all the responses and changes made, marked in highlighting in the revised manuscript. 

Response to Reviewer 1 Comments

Point 1: The authors used P < 0.05 as a cut-off for significantly differential expression change. Was this P-value corrected for multiple test? 

Response 1: Thank you for you question, we did not use it because we could only screen for a few differentially expressed genes by correcting the p-value. However, we verified some genes by RT-PCR and found that there is a high correlation between RT-PCR and RNA-seq results, at least the results of these genes are reliable.

Point 2: The age at which ovariectomy was performed should be clearly indicated. 

Response 2: Thanks for your reminder, the ovariectomy and sham-operation was performed at 3 weeks of age. (line 91)

Point 3: Line 178 is confusing, screening for what?

Response 3: Thanks for your question, we have revised according to the suggestion. (line 178)

Point 4: Are the RT-PCR data from the same samples that were used for RNA-seq analysis

Response 4: Thanks for your reminder, we have modified this issue to clearly describe the same sample used by qRT-PCR and RNA-seq. (line 129)

Point 5: (Line 234) to the than?

Response 5: Thank you for your reminder, it is a clerical error and we has been deleted it.

Point 5: What was the basis for the membrane localization of ESR1? 

Response 5: I’m sorry, it is a mistake, ESR1 should be located in the cytoplasm, and the ligand enters the cell to bind to it and play a biological role. With the following references:

Moriarty, K., .; Kim, K. H.; Bender, J. R., Minireview: estrogen receptor-mediated rapid signaling. Endocrinology 2006, 147 (12), 5557-5563.

Kim, K. H.; Bender, J. R., Rapid, estrogen receptor-mediated signaling: why is the endothelium so special? Sciences Stke Signal Transduction Knowledge Environ

Reviewer 2 Report

This is a new version of a paper already sent for revision to Genes. Unfortunately I’m not able to read the previous comments about this paper.

Page 2, line 85 local species -> local breed.

Page 3, lines 95-97: why you indicated first the age of 13, 17, 22 weeks and then 10, 14, 19 weeks?

Page 3, lines 101-103: this is a duplicate of lines 98-99 on the same page.

Page 3, line 105: why you extracted RNA only from the samples collected at 14 (or 13) weeks of age considering that you sampled this tissue at three ages?

Page 3, line 111: Illumine -> Illumina.

Page 3, line 114: Why you used Galgal4 and not Gallus_gallus-5.0 for your analyses?

Page 3, line 115: please remove the space between “ncbi.” and “nlm”.

Page 3, line 118: please confirm you reported data about “fold change” not “log2 fold change”.

Page 3, line 122: have you tried to use Cytoscape instead of DAVID? Consider this suggestion also for Results and Discussion sections.

Page 4, line 145: what does it means “(*)”?

Page 4, line 156: “male chickens” AND “ovariectomy”???? Please correct.

Page 4, paragraph 3.2: You described the results of chicken of 17 (?) weeks of age while in Methods you wrote 14 (13???) weeks of age. Please revise carefully!

Page 7, paragraph 4.1 This is a short version of the introduction, already written at the beginning of the paper.

Page 7, lines 220-222: the aim of the paper is usually written at the end of the introduction not in discussion.

Page 7: line 232: how about the 1461-108 DEGs you detected? Please describe shortly the other DEGs you found and their functions.

Page 7 and 8, lines 255-256 / 262-263: why you reported two different conclusion about the role of the two groups of genes?

Page 9, line 296: 14 and 19 weeks????

In my opinion a connection between AF and IMF is completely lacking on this manuscript. This connection was expected reading the introduction. The Authors must provide a revised version of the paper describing this connection or they must change the introduction.

Author Response

Manuscript Number: genes-410827

Identification of differentially expressed genes and pathways for abdominal fat deposition in ovariectomized and sham-operated chickens

Dear editor, 
We are grateful to reviewers for their effort, comments and valuable suggestions. We have carefully considered the comments and made several, appropriate changes to the text, point-by-point. References have been updated as well. We are confident that this revised version is superior to the original. Here are details of all the responses and changes made, marked in highlighting in the revised manuscript. 

Response to Reviewer 2 Comments

Point 1: Page 2, line 85 local species -> local breed.

Response 1: Thank you for your reminder, we have changed it. “local breed” has been replaced “local species”. (line 88)

Point 2: Page 3, lines 95-97: why you indicated first the age of 13, 17, 22 weeks and then 10, 14, 19 weeks? 

Response 2: Thank you for your question, chicken had been ovariectomy or sham-operation at 3 weeks, and13,17,22 refers to the age of weeks from birth, 10,14,19 refers to the age of weeks after ovariectomy or sham-operation, and it has been revised. (line 98)

Point 3: Page 3, lines 101-103: this is a duplicate of lines 98-99 on the same page.

Response 3: Thank you for your reminder, we have revised according to the suggestion.

Point 4: Page 3, line 105: why you extracted RNA only from the samples collected at 14 (or 13) weeks of age considering that you sampled this tissue at three ages?

Response 4: Thank you for your question, in our study, the phenotype of chickens at 10 weeks of age was not significantly different, the phenotypic differences were significant at 14 and 19 weeks of age, and the cause of phenotypic differences should be earlier, so we chose 14 weeks of age.

Point 5: Page 3, line 111: Illumine -> Illumina.

Response 5: Thank you for your reminder, we have changed it. “Illumina” has replaced “Illumine(line 111)

Point 6: Page 3, line 114: Why you used Galgal4 and not Gallus_gallus-5.0 for your analyses?

Response 6: Thank you for your question, This RNA sequencing work of this experiment was earlier than the publishtion time of Gallus-gallus 5.0 (Dec 2015)

Point 7: line 115: please remove the space between “ncbi.” and “nlm”.

Response 7: Thank you for your reminder, we have revised according to the suggestion. (line 115)

Point 8: Page 3, line 118: please confirm you reported data about “fold change” not “log2 fold change”.

Response 8: Thank you for your reminder, we have revised according to the suggestion. (line 118)

Point 9: have you tried to use Cytoscape instead of DAVID? Consider this suggestion also for Results and Discussion sections.

Response 9: Thank you for your reminder, we have revised according to the suggestion. (line 122)

Point 10: Page 4, line 145: what does it means “(*)”?

Response 10: Thank you for your reminder, it is a clerical error and we have revised according to the suggestion. (line 145)

Point 11: Page 4, line 156: “male chickens” AND “ovariectomy”???? Please correct.

Response 11: Thank you for your reminder, we have revised according to the suggestion. (line 157)

Point 12: Page 4, paragraph 3.2: You described the results of chicken of 17 (?) weeks of age while in Methods you wrote 14 (13???) weeks of age. Please revise carefully!

Response 12: Thank you for your question, we have revised according to the suggestion.

Point 13: Page 7, paragraph 4.1 This is a short version of the introduction, already written at the beginning of the paper.

Response 13: Thank you for your question, the corresponding paragraphs have been revised. (line 211)

Point 14: Page 7, lines 220-222: the aim of the paper is usually written at the end of the introduction not in discussion.

Response 14: Thank you for your question, we have revised as suggested.

Point 15: line 232: how about the 1461-108 DEGs you detected? Please describe shortly the other DEGs you found and their functions.

Response 15: Thank you for your question, we have revised according to the suggestion. line 226

Point 16: Page 7 and 8, lines 255-256 / 262-263: why you reported two different conclusion about the role of the two groups of genes?

Response 16: Thank you for your question, the meaning we want to express is not clearly clarified, both conclusions indicate that as the expression of these genes increases, abdominal fat deposition is increased. we have revised it. (line 251-252 / 258-259)

Point 17: Page 9, line 296: 14 and 19 weeks????

Response 17: Thank you for your question, the same answer as question 2 

Point 18: In my opinion a connection between AF and IMF is completely lacking on this manuscript. This connection was expected reading the introduction. The Authors must provide a revised version of the paper describing this connection or they must change the introduction.

Response 18: Thank you for your question, studies have shown that there is a genetic positive correlation between intramuscular fat and abdominal fat. (line 55)

with the following references:

1.     Zhao, G. P.; Wen, J.; Chen, J. L.; Zheng, M. Q.; Xiao-Hua, L. I., Selection Response and Estimation of the Genetic Parameters for Intramuscular Fat in a Quality Chicken Line. Acta Veterinaria Et Zootechnica Sinica 2006, 37 (9), 870-873.

2.     Chen, J. L.; Zhao, G. P.; Zheng, M. Q.; Wen, J., .; Yang, N., . Estimation of genetic parameters for contents of intramuscular fat and inosine-5'-monophosphate and carcass traits in Chinese Beijing-You chickens. Poultry Science 2008, 87 (6), 1098-104.

Round 2

Reviewer 1 Report

The revision addressed the some of the concerns raised earlier. To determine how ovariectomy affects the expression of genes in fat metabolism, the authors used currently standard widely used method, RNA-seq. Because the number of samples per treatment is small (three controls and three ovariectomized), the significance of the study diminishes, due to the potential random effects and other uncontrolled factors. The consistence of the results appears to be a concern, as the p-value was not corrected for multiple test. 

The revised conclusion reads awkward. Further revision is necessary to reflect the initial aims of the study. 

Author Response

Manuscript Number: genes-410827

Identification of differentially expressed genes and pathways for abdominal fat deposition in ovariectomized and sham-operated chickens

Dear editor,
We are grateful to reviewers for their effort, comments and valuable suggestions. We have carefully considered the comments and made several, appropriate changes to the text, point-by-point. References have been updated as well. We are confident that this revised version is superior to the original. Here are details of all the responses and changes made, marked in highlighting in the revised manuscript.

Response to Reviewer 1 Comments

Point 1: Comments and Suggestions for Authors

The revision addressed the some of the concerns raised earlier. To determine how ovariectomy affects the expression of genes in fat metabolism, the authors used currently standard widely used method, RNA-seq. Because the number of samples per treatment is small (three controls and three ovariectomized), the significance of the study diminishes, due to the potential random effects and other uncontrolled factors. The consistence of the results appears to be a concern, as the p-value was not corrected for multiple test.

Response 1: Thank you for you question. With multiple corrections for the p-value, only a few hundred differentially expressed genes were found. In order to prevent losing more important information, we selected the p-value for screening, and increased the verification by Q-PCR to ensure the accuracy of information.

In this study, abdominal fat metabolism and the related genes were taken as the focused object. A total of 1461 differentially expressed genes were screened. Among them, 108 genes related to lipid metabolism were screened. 12 interest genes were selected from 108 differentially expressed genes related to lipid metabolism for Q-PCR validation (11.11%). It was found that the expressions of 12 selected genes by Q-PCR and RNA-seq data were consistent, indicating the accuracy and reliability of transcriptome data. Moreover, the expression changes of 12 lipid metabolism genes between ovariectomized group and control group were identical with phenotype, which further explained the accuracy and reliability of transcriptome data and eliminated the possible randomness. These results indicate that the transcriptome data has high accuracy and reliability.

Point 2: The revised conclusion reads awkward. Further revision is necessary to reflect the initial aims of the study.

Response 2: Thank you for your reminder, we have revised according to the suggestion.

Reviewer 2 Report

Conclusion.

Please describe/write better whay you mean with "exporting the regulatory mechanism" and correct "some certain genes".

Results, Discussion, Conclusion.

A connection between AF and IMF is completely lacking. You don'tested the amount of IMF on the analyzed chicken so you are not able to support what you wrote on lines 60-61 and 73-75. Please reconsider this evidence on your manuscript.

Author Response

Manuscript Number: genes-410827

Identification of differentially expressed genes and pathways for abdominal fat deposition in ovariectomized and sham-operated chickens

Dear editor,
We are grateful to reviewers for their effort, comments and valuable suggestions. We have carefully considered the comments and made several, appropriate changes to the text, point-by-point. References have been updated as well. We are confident that this revised version is superior to the original. Here are details of all the responses and changes made, marked in highlighting in the revised manuscript.

Response to Reviewer 2 Comments

Point 1: Conclusion.

Please describe/write better whay you mean with "exporting the regulatory mechanism" and correct "some certain genes".

Response 1: Thank you for your reminder, we have revised the conclusion.

Point 2: Results, Discussion, Conclusion.

A connection between AF and IMF is completely lacking. You don'tested the amount of IMF on the analyzed chicken so you are not able to support what you wrote on lines 60-61 and 73-75. Please reconsider this evidence on your manuscript.

Response 1: Thank you for your reminder, we have revised according to the suggestion.